# Regulation of p53 Activity by (+)-Epiloliolide Isolated from *Ulva lactuca*

**DOI:** 10.3390/md19080450

**Published:** 2021-08-05

**Authors:** Yuheon Chung, Seula Jeong, In-Kyoung Lee, Bong-Sik Yun, Jung Sup Lee, Seungil Ro, Jong Kun Park

**Affiliations:** 1Division of Biological Science, Wonkwang University, Ikasn 54538, Korea; yhchung@wku.ac.kr (Y.C.); sla0940@wku.ac.kr (S.J.); 2Division of Biotechnoloy, Advanced Institute of Environment and Bioscience, College of Environmental and Bioresource Sciences, Jeonbuk National University, Iksan 54596, Korea; iklee@jbnu.ac.kr (I.-K.L.); bsyun@jbnu.ac.kr (B.-S.Y.); 3Department of Biomedical Science, Chosun University, Gwangju 61452, Korea; jsplee@chosun.ac.kr; 4Department of Physiology and Cell Biology, School of Medicine, University of Nevada, Reno, NV 89557, USA; sro@med.unr.edu

**Keywords:** *Ulva lactuca*, (+)-Epiloliolide, p53, UVB-induced damage

## Abstract

*Ulva lactuca (U. lactuca)* is a green alga distributed worldwide and used as a food and cosmetic material. In our previous study, we determined the effects of *U. lactuca* methanol extracts on the UVB-induced DNA repair. In the present study, we fractionated *U. lactuca* methanol extracts to identify the effective compound for the DNA repair. MTT assay demonstrated that (+)-epiloliolide showed no cytotoxicity up to 100 μM in BJ-5ta human dermal fibroblast. Upon no treatment, exposure to UVB 400 J/m^2^ decreased cell viability by 45%, whereas (+)-epiloliolide treatment for 24 h after UVB exposure significantly increased the cell viability. In GO and GESA analysis, a number of differentially expressed genes were uniquely expressed in (+)-epiloliolide treated cells, which were enriched in the p53 signaling pathway and excision repair. Immunofluorescence demonstrated that (+)-epiloliolide increased the nuclear localization of p53. Comet assay demonstrated that (+)-epiloliolide decreased tail moment increased by UVB. Western blot analysis demonstrated that (+)-epiloliolide decreased the levels of p-p53, p21, Bax, and Bim, but increased that of Bcl-2. Reverse transcription PCR (RT-PCR) demonstrated that (+)-epiloliolide decreased the levels of MMP 1, 9, and 13, but increased that of COL1A1. These results suggest that (+)-epiloliolide regulates p53 activity and has protective effects against UVB.

## 1. Introduction

Sunlight includes visible rays, infrared rays, and ultraviolet light (UV). UV is divided into A, B, and C, depending on the wavelength and increases the risk of cancer [1]. UV penetrates deeper into the skin as the wavelength increases, and UVA and UVB damage the skin [2]. UVB has a shorter wavelength than UVA, but it can cause more severe genotoxicity and cytotoxicity than UVA [3]. UVB mainly penetrates the basal cell layer of the skin epidermis but is also crosses the epidermis and reaches the papillary dermis [4]. When the skin is continuously exposed to UVB, DNA damage, such as cyclobutane pyrimidine dimer (CPD), pyrimidine (6-4) pyrimidine photoproducts (6-4PP) single strand breaks (SSBs), and 8-oxoguanin (8-OHdG) are induced because of the absorption of UVB by the nucleotide [5]. DNA damage increases the instability of DNA [6,7,8]. SSBs and 8-OHdG are removed by base excision repair (BER). In BER, DNA damage recognized by DNA glycosylase such as monofunctional and bifunctional glycosylase [9]. Those increase the AP site by removing the damaged base and recruit AP endonuclease 1 and flap endonuclease 1 [10]. CPD is induced 5–10 times more frequently by UVB than 6-4PP. CPD and 6-4PP are eliminated by nucleotide excision repair (NER) using more than 30 proteins [11,12]. NER removes bulk DNA adducts by regulating the DNA excision repair genes [13]. In NER, CPD and 6-4PP were recognized by XPC, DDB1, and DDB2 (also known as XPE) [14]. DDB complex inhibits RNA polymerase binding at DNA strand and recruits Cockayne syndrome B (CSB) and CSA to initiate the NER process [15]. NER and BER can remove single stranded DNA damage [16]. DNA excision repair genes related to NER involve in growth arrest and DNA damage α (Gadd45α), p48-XPE, and DNA polymerase, which are regulated by p53 [17].

The p53, a key protein in DNA damage response, is a transcription factor. Activated p53 changes the balance of gene expression leading to cell cycle arrest and apoptosis [18]. These changes prevent the proliferation of damaged cells. In unstressed cells, p53 is maintained at low levels by MDM2. MDM2 is part of negative-regulatory feedback loop of p53 and transcriptionally activated by p53 [19]. p53 contains transactive domain that is the same site of the MDM2 binding domain. When the DNA damage sensors such as MRN complex, RPA, and DDB1 recognize the damaged site, damage sensors phosphorylate ATM and ATR [20]. Phosphorylated ATM and ATR separate p53-MDM2 complex through p53 and MDM2 phosphorylation. p53, separated with MDM2, forms a tetramer and binds to DNA [21]. The p53 tetramer promotes DNA repair by regulating downstream genes, such as p21 and Gadd45α, to arrest the cell cycle and recruit DNA repair-related proteins [22]. However, when DNA damage is not repairable, p53 induces apoptosis by regulating pro-apoptotic proteins, such as Bax, Bak, and Bid, and antiapoptotic proteins, such as Bcl-2 and Bcl-XL [23].

*Ulva lactuca* (*U. lactuca*), also known by the common name sea lettuce, is a green alga included in the family of Ulvaceae. *U. lactuca* is distributed worldwide and inhabits rocks in the lower intertidal zone that are affected by waves. As *U. lactuca* lives in a harsh environment that is directly in contact with UV light and saltwater, it contains a large amount of flavonoids, tannins, phenols, polysaccharides, saponins, as well as vitamins [24,25,26]. Therefore, it is being studied for its potential as a feedstuff, bioenergy source, cosmetic material, and food material [27]. *U. lactuca* extracts have various biological functions, such as a protective effect against free radicals that cause skin aging [28,29]. Crude proteins extracted from *U. lactuca* have antifungal activity and reduce hypersensitivity reactions by inhibiting the proliferation of pathogenic fungi on the skin [30]. Polysaccharides present in *U. lactuca* have anti-inflammatory and anticancer effects [31,32]

(−)-Loliolide and (+)-epiloliolide, a carotenoid-derived metabolite, are classified as norisoprenoids. (−)-Loliolide was found in terrestrial such as animals and plants and marine organisms such as algae and corals. (−)-Loliolide and (+)-epiloliolide have similar structure and biological activities including anticancer, anti-inflammatory and antioxidant activities [33,34,35]. However, the two compounds are thought to differ in the binding capacity of proteins by stereoisomers.

This study was conducted to isolated and identified (+)-epiloliolide from *U. lactuca*. Additionally, we determined the gene regulation of (+)-epiloliolide using RNA sequencing, DNA repair through p53, and anti-wrinkle effects through MMPs and COL1A1 in BJ-5ta cells.

## 2. Results

### 2.1. Isolation and Purification of (+)-Epiloliolide from U. lactuca Methanol Extracts

(+)-Epiloliolide isolation was performed in three steps using *U. lactuca* methanol extracts (Figure 1). To find the effective compound for DNA repair, methanol extracts were fractionated into over 20 fractions. The fraction scheme of (+)-epiloliolide is presented in Figure 1. In the first step, four subfractions were isolated from *U. lactuca* methanol extracts using Diaion HP-20 column chromatography, and the 30% methanol fraction decreased the level of CPD more than the other fractions (data not shown). In the second step, 11 sub-fractions were isolated from the 30% methanol fraction in the first step using ODS flash column chromatography. The 30% methanol sub-fraction of second step decreased the level of CPD more than the other sub-fractions (data not shown). In the third step, three fractions were isolated from the 30% methanol subfraction in second step using Sephadex LH-20 column chromatography. The three fractions were labeled Fraction 1, 2, and 3. Fraction 2 decreased the level of CPD more than the others did (data not shown). (+)-Epiloliolide was isolated from Fraction 2 using ODS preparative HPLC (20% aq. Methanol, 3 mL/min). The amounts of (+)-epiloliolide was 1.4 mg. Additionally, the structure of (+)-epiloliolide is depicted in Figure 2A and Appendix A.

### 2.2. Cell Viability of (+)-Epiloliolide with or without UVB Irradiation

The cells exposed to 400 J/m^2^ UVB or not were treated with growth medium or medium containing various concentrations of (+)-epiloliolide for 24 h. (+)-Epiloliolide has no cytotoxicity up to 200 μM (Figure 2B). Cell viability, which decreased by approximately 55% by UVB, increased in a (+)-epiloliolide concentration-dependent manner up to 100 μM. At 100 μM of (+)-epiloliolide, cell viability increased by approximately 30% compared to UVB-exposed cells (Figure 2C). However, at 200 μM (+)-epiloliolide, cell viability decreased slightly. Thus, 100 μM (+)-epiloliolide was selected as the highest concentration for further experiments.

### 2.3. Gene Ontology (GO) and GSEA Pathway Analysis of (+)-Epiloliolide

To further characterize the related pathways underlying (+)-epiloliolide treatment, we performed GO enrichment analysis of 4377 proteins. The results showed that the biological regulation of biological processes was significantly enriched in (+)-epiloliolide treatment, with the majority of positive regulation of cell death. The molecular function and cellular component in GO also suggested that these proteins were remarkedly enriched for ‘DNA-binding transcription factor activity’ and ‘fibrillar collagen trimer’ (Figure 3A). The transcripts that were expressed in both growth medium and (+)-epiloliolide treated cells, as well as those that were uniquely expressed in (+)-epiloliolide-treated cells, were analyzed further by GO, focusing on biological process pathway. The top 5 enriched biological processes are related to p53 signaling pathway and systemic lupus erythematosus (Figure 3A). (+)-Epiloliolide decreased the mRNA levels of MDM2, SESN1, SERPINE1, and cell cycle arrest genes involve in ZMAT3, CCNG1, and CDKN1A except PMAIP1 (Figure 3B). To determine the effects of (+)-epiloliolide on p53 signaling pathway, the protein expression levels of p53, p-p53, and p21 were measured by western blotting. BJ-5ta cells were treated with various concentrations of (+)-epiloliolide for 12 h. (+)-Epiloliolide did not change the levels of p53 and p21. However, the level of p-p53 significantly increased in a concentration-dependent manner (Figure 3C,D).

### 2.4. Induction of p53 Nuclear Translocation and DNA Damage Reduction by (+)-Epiloliolide

Immunofluorescence experiments were conducted to determine the level of p53 nuclear localization in UVB-exposed cells treated with various concentrations of (+)-epiloliolide for 2 h. In the control cells, p53 foci were detected predominantly in the cytoplasm. In nontreated UVB-exposed cells, p53 foci were also detected in cytoplasm and nuclear blebbing was observed in DAPI staining. However, the treatment of (+)-epiloliolide after UVB exposure induced the translocation of p53 to nucleus and decreased nuclear blebbing. The results shown in Figure 4 suggest that (+)-epiloliolide regulates DNA repair through nuclear p53.

To determine the effects of (+)-epiloliolide on UVB-induced DNA damage, the levels of CPD were measured by IBD. CPD did not detect in DNA from cells incubated in growth medium (data not shown). After 400 J/m^2^ UVB exposure, BJ-5ta cells were treated with various concentrations of (+)-epiloliolide for 12 h. (+)-Epiloliolide decreased UVB-induced CPD level in a concentration dependent manner. 100 μM of (+)-epiloliolide decreased CPD level by approximately 45% compared to that in UVB-exposed cells (Figure 5).

Comet assay was performed to further examine the integrity of chromosomal DNA after UVB exposure. Significant DNA damages, indicated by the tail moment of the comet assay, were observed in UVB-exposed cells. Treatment with (+)-epiloliolide decreased tail moment by approximately 40% compared to that in UVB-exposed cells (Figure 6).

### 2.5. Regulation of DNA Damage Response (DDR) and Apoptosis Related Proteins by (+)-Epiloliolide

To determine the concentration dependent effects of (+)-epiloliolide on DDR and apoptosis, the protein expression levels of p-p53, p53, p21, Bax, Bim, and Bcl-2 were measured by western blotting (Figure 7, Appendix A). UVB increased p-p53 and apoptotic proteins. (+)-Epiloliolide decreased the levels of p-p53 and apoptotic proteins in a concentration-dependent manner after 12 h. At 100 μM of (+)-epiloliolide, the levels of p-p53, p21, Bim and Bax decreased by approximately 70%, 60%, 58%, and 60% and the level of Bcl-2 increased by approximately 60%, respectively, compared to those in UVB-exposed cells. (+)-Epiloliolide and ultraviolet did not change the levels of p53 at 12 h.

### 2.6. Anti-Wrinkle Effects of (+)-Epiloliolide against UVB

To determine the anti-wrinkle effects of (+)-epiloliolide in UVB-exposed cells, the levels of MMP-1, MMP-9, MMP-13 and collagen were measured by RT-PCR. The mRNA expression levels of MMP-1, MMP-9, and MMP-13 were increased by UVB exposure (Figure 8). However, MMP genes were downregulated by (+)-epiloliolide treatment in a concentration dependent manner. Compared to UVB exposure, at 100 μM of (+)-epiloliolide, the expression of MMP genes decreased by approximately 50%, 45%, and 40%. However, UVB exposure decreased the level of collagen by approximately 50% compared to the control. Treatment with (+)-epiloliolide increased the level of collagen compared to that in UVB-exposed cells. At 100 μM of (+)-epiloliolide, the level of collagen increased similar to that in the control.

## 3. Discussion

In our previous study, we demonstrated the effects of *U. lactuca* methanol extracts on DNA repair against UVB. An analysis of mineral and antioxidant contents of *U. lactuca* revealed that it contained Mg, polyphenols and anthocyanosides, which promote DNA repair [36]. Mg acts as a cofactor for UV-induced excision repair enzymes, such as AP endonuclease, DNA polymerase β, XPD, and XPG [37]. Polyphenols and anthocyanosides, which are antioxidants, increase the activity of nuclear factor erythroid-related factor-2 (Nrf-2) [38]. Activated Nrf-2 removes ROS through the transactivation of downstream genes, such as catalase and superoxide dismutase [39]. However, antioxidants cannot directly remove DNA damage. Therefore, we attempted to isolate an active compound that can directly affect the removal of DNA damage. Various fractions were isolated from *U. lactuca* methanol extracts, and the reduction of CPD was determined. As a result, we isolated (+)-epiloliolide as the effective compound for DNA repair from *U. lactuca* methanol extracts.

(+)-Epiloliolide has a better anticancer effect than (−)-loliolide [39]. Most studies have reported on the functions of (−)-loliolide. (−)-Loliolide decreased oxidative stress induced DNA damage and p53 [38]. In Figure 3, KEGG pathway analysis showed that (+)-epiloliolide regulated the expression of p53 signaling pathway related genes. (+)-Epiloliolide decreased the gene expression of apoptosis and MDM2. These results suggest that (+)-epiloliolide activates p53 through inhibition of MDM2 mRNA.

UVB crosses the epidermis and reaches the upper dermis. Human dermal fibroblasts (HDFs), continuously exposed to UVB, are considered as a versatile model to study skin photoaging since these cells are easily cultured and responsive to various aging related stimuli [40,41]. Senescent fibroblasts by acute and chronic exposure to UVB overexpress metalloproteases and accumulation of these cells in the dermis contributes to overall skin photoaging [42]. Additionally, exposure to UVB in HDFs is known to increase the overall levels of p53 and to phosphorylate this protein strongly [43]. p53 regulates the expression of DNA repair mechanisms related genes and maintains genome stability [17,19]. UVB-induced DNA damage activates and phosphorylates DNA damage sensor proteins, such as ATM, DNA -PK, ATR, and Chk2 [12]. Phosphorylated p53 activated by ATM inhibits cell proliferation through the transactivation of p21 and Gadd45-α [17]. p21 inhibits cyclin-dependent kinase and arrests the cell cycle from G1 to S [44]. p53 also regulates the expression of ER-related genes, such as Gadd45-α, p48-XPE, and DNA polymerase [45]. Gadd45-α binds to UVB-induced DNA damage lesions and recruits DNA repair proteins [46]. Additionally, Gadd45-α interacts with proliferating cell nuclear antigen (PCNA) and p21. PCNA cleaves the 3′ nucleotide by forming a pre-incision NER complex with XPG. DNA polymerase ε and δ fill the DNA gap induced by the excision of NER [47].

In cells exposed to UVB, treatment with (+)-epiloliolide increased the translocation of p53 to nucleus at 2 h (Figure 4) and decreased the protein levels of p-p53 and p21 at 12 h (Figure 7). These results suggest that the increased translocation of p53 to nucleus at early time points after treatment of (+)-epiloliolide enhances DDR, which results from the sequential actions of p-p53 and p21. The early upregulation of DDR by (+)-epiloliolide decreases the levels of UVB-induced DNA damage (Figure 5 and Figure 6) and pro-apoptotic proteins such as Bim and Bax (Figure 7) and increase cell viability (Figure 2) at later time points.

Genomic instability induced by DNA damage cause cell cycle arrest and replication arrest. If the DNA damages are not properly repaired, prolonged cell cycle arrest induces cell senescence or cell death. Thus, DNA damages are the onset of the aging process. The age-associated decline of DNA repair increases imbalance between proliferation and apoptosis. This imbalance can accelerate skin wrinkle through increase of MMP proteins and degradation of collagen protein. The early upregulation of DDR by (+)-epiloliolide would decrease the levels of MMPs and increase the level of COL1A1 at the later time points (Figure 8).

## 4. Materials and Methods

### 4.1. Algal Materials

*U. lactuca* was collected from the coast of Gijang, Korea. Fresh *U. lactuca* was washed three times with tap water to remove salt and sand, and then carefully rinsed with distilled water and dried in the shade. Dried *U. lactuca* was ground into a powder.

### 4.2. Isolation of (+)-Epiloliolide from U. lactuca

*U. lactuca* powder (500× *g*) was extracted twice at 40 °C for 4 h with 70% methanol (2 L) using an extractor (KSNP B1130-240L, Kyungseo, Incheon, Korea). The methanolic extract was concentrated under reduced pressure, followed by adsorption onto Diaion HP-20 resin (Mitsubishi Chemical Co., Tokyo, Japan) with stirring for 3 h. After filling the adsorbed resin into the column, the column was washed with distilled water and sequentially eluted with 2 L of 30%, 70%, and 100% methanol. The 30% methanol fraction was concentrated and sequentially eluted with 0–100% methanol using ODS flash column chromatography (Cosmosil 75C18-prep, Nacalai Tesque, Tokyo, Japan). Methanol (30%) fraction was concentrated and sequentially eluted with 50% methanol using Sephadex LH-20 column chromatography (GE Healthcare Life Sciences, Uppsala, Sweden). The three fractions were eluted using Sephadex LH-20 column chromatography.

### 4.3. Identification of (+)-Epiloliolide from U. lactuca

To determine the chemical composition of Fraction 2, the high-resolution mass spectrum was determined using an electrospray ionization (ESI)-QTRAP-3200 mass spectrometer (ESI-MS; Applied Biosystems, Forster, CA, USA). UV and IR spectra were recorded on Shimadzu UV-300 and FT-IR Equinox 55 spectrophotometers, respectively. Nuclear magnetic resonance (NMR) spectra were obtained on a JEOL JNM-ECA600, 600 MHz FT-NMR spectrometers at 600 MHz for ^1^H NMR, and at 150 MHz for ^13^C NMR in CD_3_OD. Chemical shifts are given in ppm with tetramethylsilane as the internal standard. Fraction 2 afforded a colorless gum which was identified as (+)-epiloliolide according to ESI-MS and NMR data: ESI-MS *m*/*z* 196.8 [M + H] ^+^, ^1^H-NMR (300 MHz, CD_3_OD) δ: 1.28 (3H, s, axial, H-10), 1.28 (1H, dd, J = 9.8 and 9.8 Hz, H-5), 1.31 (3H, s, H-9), 1.41 (1H, dd, J = 11.7 and 11.7 Hz, H-7), 1.58 (3H, s, axial, H-8), 2.00 (1H, ddd, J = 2.2, 4.3 and 13.0 Hz, H-5), 2.47 (ddd, J = 2.2, 3.9 and 11.7 Hz, H-7), 4.09 (1H, m, H-6), 5.77 (1H, s, H-3); ^13^C-NMR (75.5 MHz, CD_3_OD) δ: 25.3 (C-10), 25.8 (C-9), 30.3 (C-8), 36.2 (C-4), 48.7 (C-7), 50.6 (C-5), 65.3 (C-6), 88.6 (C-7a), 113.7 (C-3), 174.0 (C-2), 183.9 (C-3a).

### 4.4. Cell Culture and Cell Viability Assay

BJ-5ta, a human dermal fibroblast cell line, was purchased from ATCC (Rockville, MD, USA). The cells were grown in culture dishes in a humidified atmosphere containing 5% CO_2_ at 37 °C. The culture medium was DMEM (Gibco, Grand Island, NY, USA) and media 199 (Gibco, Grand Island, NY, USA) (4:1) supplemented with 10% fetal bovine serum, penicillin (100 U/mL), and streptomycin (100 μM). The pH of the medium was adjusted to 7.2–7.4 with 10 mM HEPES buffer. The cells were maintained in the exponential phase by subculturing using 0.025% trypsin-EDTA. The dose of UV irradiation was calibrated using a UV radiometer (UVP Inc., Upland CA, USA). For in vitro studies, cells were seeded 24 h prior to irradiation and washed once with phosphate-buffered saline (PBS).

Cell viability was measured using a 3-[4,5-dimethylthiazol-2-yl]-2,5-diphenyl tetrazolium bromide (MTT) assay. Briefly, the cells were cultured in 96-well plates, exposed to UVB, and incubated in a medium containing various concentrations of (+)-epiloliolide for 24 h. Then, 10 μL of MTT solution (5 mg/mL in PBS) was added and the cells were incubated for 4 h at 37 °C. Finally, DMSO was added to each well, and the absorbance was measured at 570 nm using a microplate reader.

### 4.5. Immunofluorescence

For immunofluorescence, BJ-5ta cells were seeded on coverslips coated with 0.1% gelatin in 24 well plate. The cells were washed once with PBS, exposed to UVB, and post- incubated with various concentrations of (+)-epiloliolide for 2 h. The cells were briefly washed with PBS and fixed with 4% formaldehyde for 3 min at room temperature. After 3 min, the cells were washed thrice with PBS and permeabilized with 5% Triton X-100 for 1 h. Cells were blocked for 1 h in 0.5% BSA at room temperature and incubated with mouse anti-p53 (1:1000, Cell Signaling, Beverly, MA, USA) over 12 h at 4 °C. The cells were washed twice with PBS and incubated with Alexa Flour 488 goat anti-mouse (1:1000, Bethyl, Montgomery, TX, USA) over 12 h at 4 °C. The cells were counterstained with DAPI (Sigma-Aldrich, St. Louis, MO, USA). The coverslips were then imaged using the Zeiss confocal microscope (Axioskop 2 plus).

### 4.6. Immunodot Blot (IDB)

DNA samples were extracted from cells exposed to UVB and post-incubated with (+)-epiloliolide. Heat-denatured DNA (1 μg) was loaded onto a positively charged polyvinylidene fluoride membrane (0.45 μm; Millipore, Schwalbach, Germany). After blotting, the dots were rinsed twice with 100 μL TBS containing 0.05% Tween 20 (TBS-T) and incubated with anti-CPD (Cosmo Bio, Tokyo, Japan) antibody (1:5000 dilution in 2% skim milk) at 37 °C for over 12 h. The blots were then washed with TBS-T and incubated with peroxidase conjugated secondary antibody for 2 h. Peroxidase activity was detected using the enhanced blotting detection system, and the membranes were immediately exposed to ChemiDoc (Bio-Rad, Hercules, CA, USA) at different time points and various concentrations.

### 4.7. Comet Assay

Cells were mixed with 0.5% of low melting agarose and were layered as microgels on 1.5% agarose pre-coated slide glass. Slides were lysed in 2.5 M NaCl, 100 mM EDTA, 10 mM Tris HCl at 10 °C for 20 min and electrophoresed for 30 min at 23 V/300 mA. The results were visualized under a fluorescent microscope after staining of gels with ethidium bromide. The results were quantified by open comet.

### 4.8. Western Blotting

Total proteins were extracted using RIPA buffer (50 mM Tris-HCl pH 8, 150 mM NaCl, 0.5% triton X-100, and 0.5% sodium deoxycholate) supplemented with phosphatase inhibitor cocktail. Protein contents quantification was determined by Bradford assay. 30 μg of total protein were resolved by SDS-PAGE using E-buffer and transferred onto PVDF membrane at 100 V for 1 h. After transfer, membranes were washed with 0.05% TBS-T and blocked with 5% skim milk. The membrane was then incubated with primary antibodies (diluted to 1:500 in the blocking buffer) overnight at 4 °C. The primary antibodies were used as follows: antibodies raised against p53, p-p53, p21, Bax, Bim, Bcl-2, and β-actin. The membrane was washed two times with TBS-T and incubated with anti-rabbit or mouse IgG coupled to HRP (diluted to 1:4000 in the blocking buffer) as a secondary antibody for 2 h at RT. Chemiluminescence detection was performed with ECL kit using ChemiDoc (Bio-Rad, Hercules, CA, USA).

### 4.9. Reverse Transcription PCR (RT-PCR)

Total RNA was extracted from cells exposed to UVB and post-incubated with (+)-epiloliolide using TRIzol reagent (Invitrogen, Carlsbad, CA, USA) according to the manufacturer’s instructions, and was reverse-transcribed using the First-Strand cDNA Synthesis Kit (iNtRON, Seongnam, Korea). RT-PCR was carried out in a 10 μL reaction mixture that contained 1 μL cDNA as template, 1 pmol/μL specific oligonucleotide primer pair, and 4 μL Taq mixture containing 0.5 U Taq DNA polymerase.

### 4.10. Statistical Analysis

All data obtained were compared using Student’s *t*-test to determine the statistical significance between groups. Values are expressed as the mean ± SD of at least three independent experiments.

## Figures and Tables

**Figure 1 marinedrugs-19-00450-f001:**
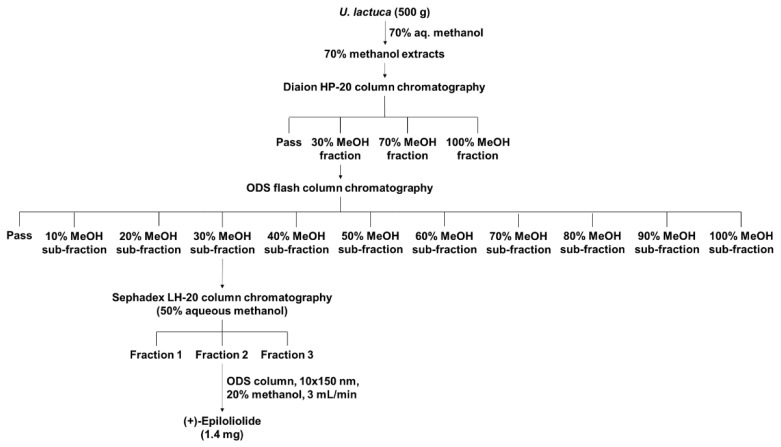
Isolation scheme of (+)-epiloliolide.

**Figure 2 marinedrugs-19-00450-f002:**
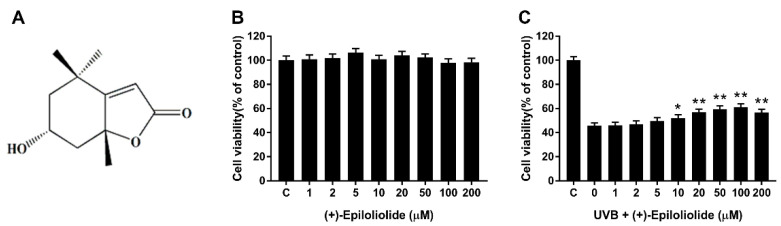
Effects of various concentrations of (+)-epiloliolide on cell viability in UVB-exposed BJ-5ta cells. (**A**) Structure of (+)-epiloliolide. Cells exposed to 400 J/m^2^ UVB (**C**) or not (**B**) were post-incubated in growth medium (GM) or medium containing various concentrations of (+)-epiloliolide for 24 h. The cell viability was determined by MTT assay. Data represent the mean values of at least three independent experiments ± SD. * *p* < 0.05 and ** *p* < 0.01 versus the non-treated (0 group) UVB-exposed group.

**Figure 3 marinedrugs-19-00450-f003:**
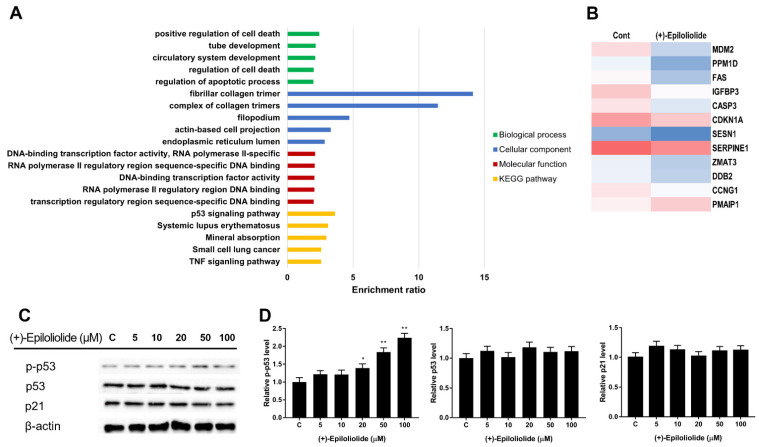
GO enrichment analysis of proteins specific in (+)-epiloliolide treated BJ-t5a cells. (**A**) The top 10 enrichment of GO and KEGG pathway of (+)-epiloliolide treatment. (**B**) Heatmap representation of the expression levels of DEGs associated with p53 signaling pathway. The color gradient from blue to red corresponds to increasing gene expression. (**C**,**D**) Cells incubated in GM or medium containing various concentrations of (+)-epiloliolide for 12 h. The levels of p-p53, p53, and p21 were determined using western blotting. Data represent the mean values of at least three independent experiments ± SD. * *p* < 0.05 and ** *p* < 0.01 versus the control group.

**Figure 4 marinedrugs-19-00450-f004:**
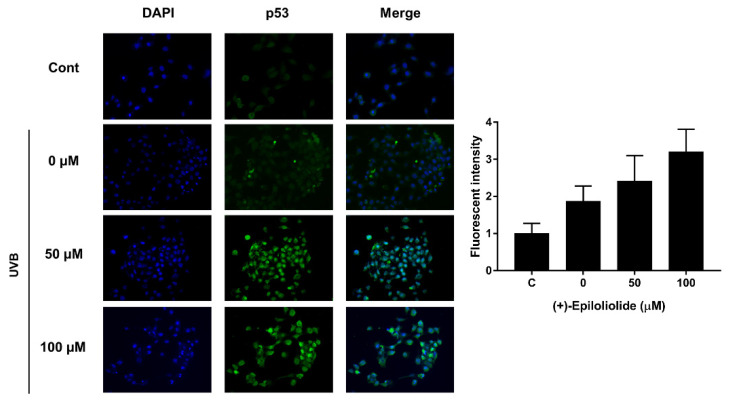
Representative immunofluorescence images showing p53 foci in UVB-exposed BJ-5ta cells. Cells exposed to 400 J/m^2^ UVB were post-incubated in GM or medium containing various concentrations of (+)-epiloliolide for 2 h. Formaldehyde-fixed BJ-5ta cells were directly stained using anti-p53 and anti-mouse conjugated to Alexa Fluor 488 to visualize p53. Stained nuclei were counterstained with DAPI.

**Figure 5 marinedrugs-19-00450-f005:**
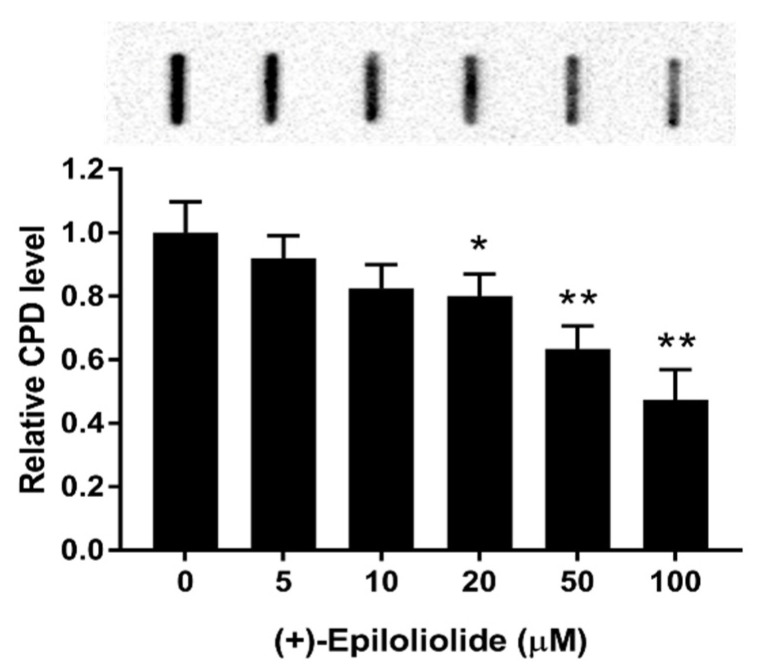
Effects of (+)-epiloliolide on the level of CPD in UVB-exposed BJ-5ta cells. Cells exposed to 400 J/m^2^ UVB were post-incubated in GM or medium containing various concentrations of (+)-epiloliolide for 12 h. Data represent the mean values of at least three independent experiments ± SD. * *p* < 0.05 and ** *p* < 0.01 versus the non-treated (0 group) UVB-exposed group.

**Figure 6 marinedrugs-19-00450-f006:**
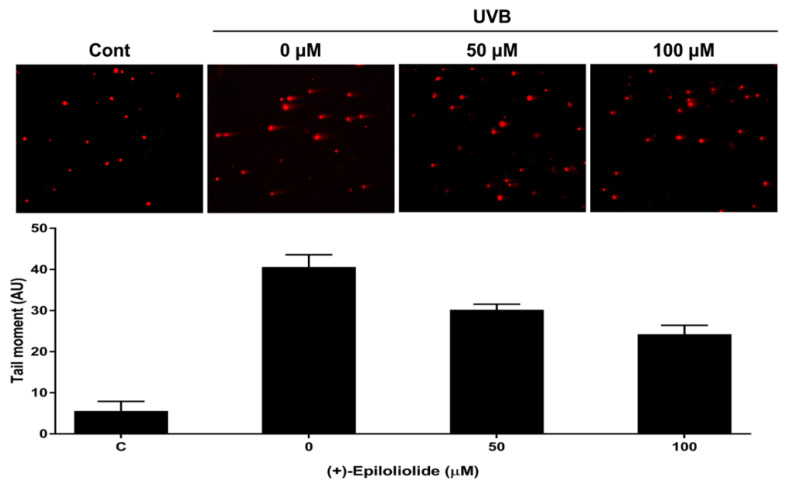
Alkaline comet assay of UVB-exposed BJ-5ta cells, treated for 2 h with various concentrations of (+)-epiloliolide. Comet images of the BJ-5ta cells stained with EtBr. Comet assay data were analyzed by specialized ImageJ software OpenComet representing the parameter tail moment. The parameter essentially represents the product of the percentage of total DNA in the tail and the distance between the centers of the mass of head and tail regions [Tail moment = (tail mean−head mean) x% of DNA in the tail].

**Figure 7 marinedrugs-19-00450-f007:**
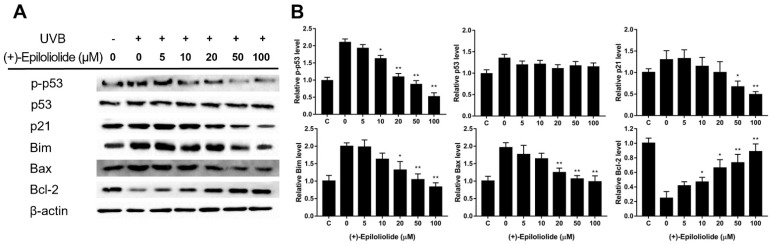
Effects of various concentrations of (+)-epiloliolide on DDR and apoptotic activities in UVB-exposed BJ-5ta cells. (**A**) Cells exposed to 400 J/m^2^ UVB were post-incubated in GM or medium containing various concentrations of (+)-epiloliolide for 12 h. The levels of p-p53, p53, p21, Bax, Bim, and Bcl-2 were determined using western blotting. (**B**) Quantitative analysis of protein expression levels of Western blot data presented in (**A**). Data represent the mean values of at least three independent experiments ± SD. * *p* < 0.05 and ** *p* < 0.01 versus the non-treated (0 group) UVB-exposed group.

**Figure 8 marinedrugs-19-00450-f008:**
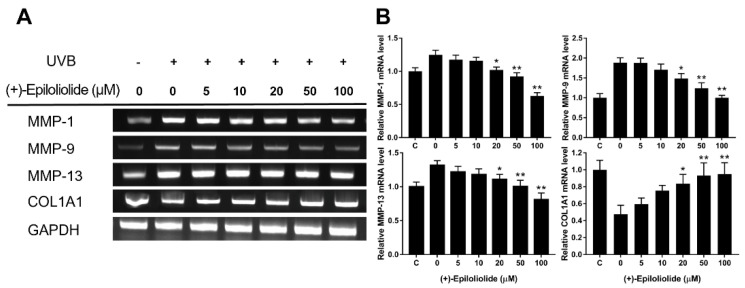
Effects of various concentrations of (+)-epiloliolide on anti-wrinkle activities in UVB-exposed BJ-5ta cells. (**A**) Cells exposed to 400 J/m^2^ UVB were post-incubated in GM or medium containing various concentrations of (+)-epiloliolide for 12 h. The levels of MMP-1, MMP-9, MMP-13, and collagen were determined by RT-PCR. (**B**) Quantitative analysis of mRNA expression levels of RT-PCR data presented in (**A**). Data represent the mean values of at least three independent experiments ± SD. * *p* < 0.05 and ** *p* < 0.01 versus the non-treated (0 group) UVB-exposed group.

## Data Availability

Data are contained within the article or Appendix A.

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
