# Peer review of "Regulation of p53 Activity by (+)-Epiloliolide Isolated from Ulva lactuca"

_marinedrugs, 2021, doi:10.3390/md19080450_

Round 1

Reviewer 1 Report

Dear Authors, 

Your manuscript has some relevant data but needs to be improved, as follows:

i) In addition to some corrections of inaccuracies (highlighted in the attached document) the manuscript should be reviewed by a native English for punctuation errors and correction of sentences.

ii) Data from the spectroscopic analysis of the isolated compound must be included in the manuscript or available at Supporting Information.

iii) Discussion is too poor and needs to be improved to support better the obtained results.

Reviewer 2 Report

Abstract: name the component (“effective compound”) in the extract as the –epiloleolide, and in the results or discussion explain why it is stated as the most effective (and name the other components it is being compared with).

Manuscript: Explain the reason for the use of fibroblasts when UVB reaches only the epidermis/keratinocytes.

Introduction: edit the last paragraph to specify all the targets, including the MMP/collagen

Methods/Results/Discussion: discuss the reasons for not having experiments with the –epiloleolide alone (without UVB) in Figures 4-8.

Figure 1: use a different symbol to show if there were significant differences between the –epiloleolide alone (Fig 1B) and –epiloleolide + UVB (Fig 1C) at each of the concentrations of the –epiloleolide

Figure 5: discuss the reason for the lack of the control (C). Specify the reason for showing –loliotide, and move the background information on it from the discussion to the introduction.

Explain the increase in nuclear p53 localization (Fig 4) versus the inhibition of p-p53 (Fig 7)

Author Response

Dear Reviewer 2,

I greatly appreciate your patience and generosity during handling my manuscript (marinedrugs-1297990).

The critics were very helpful for my research. According to your comments, I undertook several experiments and the data were included in the revised manuscript. In my revised manuscript, Discussion part was rewritten to provide more detailed framework of the study. Figures and Figure legends were sufficiently corrected according to your comments. To exclude typographic errors, my revised manuscript was carefully examined at least three times by three scientists.

I would like to explain how I revised the manuscript according to the critics.

Response to the critics of Reviewer 2.

1. Abstract: (+)-Epiloliolide was effective compound (not most effective) for DNA repair from lactuca methanol extracts. We fractionated U. lactuca methanol extracts into over 20 fractions and the reduction of CPD was determined to select effective fraction for DNA repair

2. Manuscript: According to the comments, the reason for use of fibroblasts was explained in Discussion.

3. Introduction: Last paragraph was corrected according to the comments.

4. Methods/Results/Discussion: According to the comments, western blot data for (+)-epiloliolide alone (without UVB) was given in Figure 2C, 2D.

5. Figure 2: According to the comments, X axis name of graph was corrected in Figure 2C.

6. Figure 5: According to the comments, the reason for the lack of control was explained in Results. CPD data for (-)-loliolide was deleted from Figure 5. The background information of (-)-loliolide and (+)-epiloliolide was moved from the Discussion to Introduction.

7. The increase in nuclear p53 localization (Figure 4) versus the inhibition of p-p53 (Figure 7) was explained in Discussion.

Thank you for your assistance and I look forward to hearing a good news from you very soon.

Sincerely yours,

Jong Kun Park, Ph, D,

Round 2

Reviewer 1 Report

Dear Authors,

I have read again your manuscript and your response to my comments was adequate. Before publication, please, consider some small corrections I have highlighted in the attached version v4.

Best regards

Author Response

Dear Reviewer 1,

I greatly appreciate your patience and generosity during handling my manuscript (marinedrugs-1297990).

I would like to explain how I revised the manuscript according to the critics.

Response to the critics of Reviewer 1.

  1. According to the comments, the inaccuracies were corrected in manuscript.

Thank you for your assistance and I look forward to hearing a good news from you very soon.

Sincerely yours,

Jong Kun Park, Ph, D,
